# Zero-Valent Iron Nanoparticles Remediate Nickel-Contaminated Aqueous Solutions and Biosolids-Amended Agricultural Soil

**DOI:** 10.3390/ma14102655

**Published:** 2021-05-19

**Authors:** Ahmed M. Mahdy, Tiequan Zhang, Zhi-Qing Lin, Nieven O. Fathi, Rasha M. Badr Eldin

**Affiliations:** 1Department of Soil and Water Sciences, Faculty of Agriculture, Alexandria University, Alexandria 21568, Egypt; rasha_792000@yahoo.com or; 2Harrow Research and Development Centre, Agriculture and Agri-Food Canada, Harrow, ON N0R 1G0, Canada; tiequan.zhang@canada.ca; 3Department of Environmental Sciences & Department of Biological Sciences, Southern Illinois University—Edwardsville, Edwardsville, IL 62026-1099, USA; zhlin@siue.edu; 4Soil Salinity & Alkalinity Laboratory, Ministry of Agriculture, Alexandria 21568, Egypt; nieven74@yahoo.com

**Keywords:** Ni^+2^, sorption, nanoparticles, biosolids, nanoscale zero-valent iron (nZVI)

## Abstract

Nickel (Ni^+2^) accumulation in wastewater treatment sludge poses a potential environmental risk with biosolids-land application. An incubation experiment was conducted to evaluate the effect of nanoparticles of zero-valent iron (nZVI) on Ni^+2^ sorption in biosolids-treated agricultural soils. Two application rates of biosolids (0, 5%, *w*/*w*) and four treatment levels (0, 1, 5, and 10 g/kg) of nZVI were examined, either separately or interactively. The results of this study showed significant differences in Ni^+2^ sorption capacity between different nZVI treatments. The initial Ni^+2^ concentration in biosolids-amended soil significantly affected Ni sorption in the soil treated with nZVI. The “H-shape” of sorption isotherm in nZVI-treated soil reflects strong interaction between the Ni concentration and the nZVI treatment, while the C-shape of sorption isotherm in biosolids-amended soil without the nZVI treatment indicates intermediate affinity for Ni^+2^ sorption. Nickel retention in soil was increased with the increase of nZVI levels. The removal efficiency of Ni^+2^ by nZVI from solution was increased with the increase of pH from 5 to 11 and reached a maximum of 99.56% at pH 11 and nZVI treatment of 10 g/kg. The Ni^+2^ desorption rate decreased from 92 to 7, 4, and 1% with increasing nZVI treatment levels from 0 to 1, 5, and 10 g/kg, respectively, with a soil Ni^+2^ concentration of 50 mg/L. The maximum adsorption capacity (𝑞_max_) of 10 g/kg nZVI-treated soil was 333.3 mg/g, which was much higher than those from the other treatments of 0 (5 mg/g), 1 (25 mg/g), and 5 g/kg (125 mg/g). The underlying mechanism for Ni^+2^ immobilization using nZVI in an aquatic environment is controlled by a sorption process, reduction of metal ion to zero-valent metal, as well as (co)precipitation. Moreover, increasing the nZVI treatment level in biosolids-amended soil significantly decreased bioavailable Ni^+2^ concentrations in the soil.

## 1. Introduction

Soil contamination with heavy metals, such as nickel, is a significant and global problem [1]. Anthropogenic activities, such as mining, tanning, textile, metallurgical [2], and electroplating processes, have resulted in severe soil and water nickel contamination. It has also been reported that high contents of toxic Ni^+2^ were found in municipal wastewater treatment sludge. Thus, the land application of biosolids may result in toxic Ni^+2^ contamination in biosolids-amended agricultural soil and also in land surface runoff.

In situ remediation of Ni^+2^-contaminated soil and water can be achieved through different physical, chemical, and biological processes. For example, Ni^2+^ can be removed from contaminated soils and waters by chemical adsorption and precipitation using nanoscale zero-valent iron (nZVI) [3,4,5]. Chemical reduction of Ni^2+^ to Ni^0^ will also reduce bioavailability of Ni^+2^ in contaminated soils [6,7]. Nanoscale particles of <100 nm in diameter [8,9,10] have a large reactive surface area and adsorbing sites, in comparison to bulk or micro-sized adsorbent materials of the same chemical composition. Nanoscale zero-valent iron (nZVI) particles have been used previously as active adsorbing materials for the remediation of toxic metal contamination [11,12], including As(V, III), Cd(II), Cr(VI), Cu(II), and Ni(II). The in situ remediation technique utilizing nZVI has many advantages, such as less toxic, non destructive, cost- and time-saving [13,14,15]. We hypothesized that nZVI particles could also significantly reduce Ni bioavailability in agricultural soils amended with Ni^+2^-contaminated biosolids.

The specific objectives of this study were to (1) examine the factors that influence the Ni sorption on nZVI in biosolids-amended agricultural soils, (2) evaluate the retention and removal efficiencies of nZVI for Ni^+2^ in soil and water, and (3) determine the possible changes in Ni fractions in the biosolids-amended soil.

## 2. Materials and Methods

### 2.1. Sampling and Preparation of Soil, Biosolids, and nZVI

A sandy loam soil (*Typic xeropsamments)* from El-Bostan, El-Behaira governorate, Egypt was selected for this incubation experiment. Surface soils (0–30 cm) were sampled, air-dried, ground, and sieved through 2 mm. Biosolids (i.e., treated municipal wastewater treatment sludge) were collected from the Municipal Wastewater Treatment Plant in Alexandria (Station No.26), Egypt, and air-dried, ground, and sieved (<2 mm) [16]. The nanoscale zero-valent iron particles of 40–45 nm in diameter were purchased from Sigma (St. Louis, MO, USA). The general physiochemical properties of studied soil and biosolids were determined according to the standard methods [17] and compiled in Table 1.

### 2.2. Incubation Experiment

A laboratory incubation experiment was conducted to stabilize the experimental soils being treated with biosolids and/or nZVI for equilibrium, including two biosolids application rates of 0 and 5% (*w*/*w*) and four nZVI treatment levels of 0, 1, 5, and 10 g/kg, either separately or combined. Concentration of Ni in the collected biosolids was 6.23 mg Ni/kg. The soil mixes were put in different jars, moistened, and equilibrated at 80% of the water holding capacity at 25 ± 2 °C for 6 days. This incubation experiment was carried out in a CRD design with three replicates. At the end of the incubation period, the soil sampling was carried out and prepared for future analyses.

### 2.3. Nickel Sorption and Removal Efficiency

The soil Ni^+2^ retention capacity was determined for each treatment and control soil using traditional batch equilibration technique [18]. Each soil sample (0.2g) was mixed with 20 ml of 0.01 *M* KCl solution containing Ni^+2^ concentrations of 0, 50, 100, 200, 400, or 800 mg/L and shaken on a shaker [120 rpm] for the equilibrium time of 24 hrs. The soil solutions were centrifuged at 5000 ×g for 15 minutes. Concentrations of Ni^+2^ in extraction solutions were measured.

The amount of sorbed Ni^+2^ with soil was calculated:q_e_ = V (C_0_ − C_e_)/m(1)
where the amount of sorbed Ni^+2^ per gram of soil is referred to as q_e_, V refers to the volume of solution, m is the weight of soil, C_0_ is the initial Ni^+2^ concentration, and C_e_ is the Ni^+2^ concentrations at equilibrium.

The removal efficiency of Ni^+2^ from solution was determined from the following formula:Efficiency (%) = (C_i_ − C_e_)/C_i_ × 100(2)
where C_i_ and C_e_ are the initial and equilibrium concentrations of Ni^+2^, respectively.

### 2.4. Adsorption Isotherm Models

Langmuir and Freundlich, the most popular models, were selected to assess the adsorption efficiency of nZVI. The Langmuir sorption isotherm model has been widely used to describe the Ni^+2^ sorption process between active sites in soil and the number of Ni^+2^ molecules sorbed on active sites in soil. The Langmuir model describes the Ni^+2^ sorption process by only one layer sorption. The Langmuir isotherm model can be described as [19]:q_e_ = q_max_(K_L_ C_e_/1 + K_L_C_e_)(3)

On the contrary, the Freundlich sorption isotherm model can describe the Ni^+2^ sorption process by multi-molecular layer adsorption. The Freundlich model can be expressed as [20]:q_e_ = K_F_C_e_^1/n^(4)
where C_e_ is the Ni^+2^ concentration (mg/L) at equilibrium; q_e_ is the sorption capacity (mg/g) at equilibrium; q_max_ is the maximum sorption capacity (mg/g); K_L_ is the sorption constant of Langmuir (L/mg) at equilibrium; K_F_ and n are Freundlich isotherm constants related to the sorption capacity.

### 2.5. Nickel Sorption Kinetics

The batch method described by Amacher et al. [21] was used to assess the Ni kinetic retention with the highest Ni^+2^ treatment level of 800 mg/L and to quantify adsorption isotherms for Ni^+2^ in control soil and the soils amended with biosolids and treated with nZVI particles at 25 °C at different time points of 5, 30, 60, 120, 240, 480, 960, and 1440 min.

### 2.6. Kinetics Modeling

To understand the effect of shaking extraction time on the sorption process of Ni^+2^, two common (first-order and power function) kinetic models were applied to the sorption data. The two equations are as follows [22,23]:First-order rate model In (qo − q) = a − ka t(5)
Power function model q = ka Co t^1/m^(6)
where:

q = the amount of sorbed Ni per gram of soil mix in a specific time t

K_a_ = the coefficient of sorption diffusion rate (μg/ g min)

C_o_ = the initial Ni^+2^ concentration (mg/ L)

t = reaction time (min)

k_a_ = the coefficient of sorption rate (min^−1^)

1/m = a constant

q_o_ = the amount of sorbed Ni at equilibrium

### 2.7. Nickel Fractions in Biosolid-Amended Soil

Fractions of nickel in the soil amended with biosolids was conducted for both nZVI-treated and un-treated samples utilizing the successive extraction procedure according to Tessier 1979 [24]. The extraction procedure differentiated soil nickel into five fractions: exchangeable, bound to carbonates, bound to Fe–Mn oxides, bound to organic matter (OM), and residual fractions (RS). Triplicates of all analyses were carried out, and Ni concentrations in filtrates were measured by inductively coupled plasma (ICP) and calculated according to the weight of soil–biosolids–nZVI mix.

### 2.8. Nickel Release

Desorption behavior of Ni^+2^-loaded nZVI was studied to test its stability. The amount of de-ionized water (10.0 mL) was added to 1.0 g Ni-loaded nZVI and shaken for 24 hrs. After shaking and centrifugation, the suspension solutions were then filtered. The Ni concentration was measured in filtrates by ICP.

### 2.9. Packed-Column Experiment

Under continuous flow conditions, the potential of nZVI to remove Ni^+2^ from contaminated water was studied using packed-column. The column was made of clear acrylic (Ø 2 cm, 35 cm) and filled with 20 g coarse sand; 1, 5, or 10 g nZVI treated and biosolid-amended soil; and 10 g fine sand. A total of ten columns, including two control columns (control and biosolids-amended soil without nZVI treatment), were constructed to evaluate the performance of the columns at four doses:sand ratios. Industrial wastewater effluents (100 mL) spiked with Ni^+2^ of 56.23 mg/L was continuously pumped into the columns, and effluent concentrations were measured with performance time. The performance of the packed bed column under different conditions was evaluated from breakthrough curves, and the optimal application rate of nZVI for best pollutant stabilization was determined.

The five pore volumes of the injection solution were then pumped into the column in a downward direction at 0.5 ml/min flow rate. Effluent samples were collected, and Ni^+2^ concentrations were determined.

The capacity of the Ni^+2^ removal column for nickel sorption (q) was calculated:q = (C_0_V_0_ − ∑C_n_ V_n_)/m(7)
where q is the amount of Ni^+2^ sorbed per gram of nZVI (mg/g), C_o_ is the initial concentration of Ni (mg/L), V_o_ is the total volume of the influent solution (L), C_n_ is the concentration of Ni in sample n (mg/L), V_n_ is the volume of sample n (L), and m is the amount of nZVI (g). After finishing the experiment, the nickel-loaded nZVI will be regenerated and can be reused for another number of cycles [18,25].

## 3. Results and Discussion

### 3.1. Characterization of nZVI Particles

The X-ray diffraction (XRD) patterns of nZVI showed apparent peaks 2θ = 45° and 64.88°, indicating the presence of metallic iron products [25]. The characteristic peaks observed at 2 θ of 24° and 37° could be attributed to the presence of maghemite (Fe_2_O_3_) and magnetite (Fe_3_O_4_), respectively.

The morphology of nZVI particles was analyzed by scanning electron microscopy (SEM) detection, and the results clearly showed that the representative single particle size dimension lies in the range 1–100 nm (nanostructure), the nZVI particles are spherical with an average size of 42–45 nm, and the particles are uniformly distributed. The energy dispersive Xray (EDX) elemental analysis of nZVI confirmed the presence of Fe in the sample by appearance of Fe (56.54%) peak in nZVI sample. Moreover, the size distribution of Zetasizer analysis refers to size average 43 nm (<100 nm).

The specific surface area (SSA) of nZVI (211.53 m^2^/g) was quite high. Indeed, the high SSA could supply nZVI with highly reactive sites for Ni^+2^ sorption. These results coincided with the results of Mahdy et al. [18] and Dongsheng et al. [26]. The electric potential across the layer in the electric double-layer in soil is called “zeta potential”. The zeta potential value reflects the quantity of charges absorbed by the solid [27,28,29]. Based on the above concept, the zeta potential characteristics of nZVI were studied. The results of current study showed that the zeta potential of nZVI was 27.36 mV, and this value of zeta potential indicated that particles of nZVI have more charges and consequently have more sorptive capacity for pollutants. Therefore, the nZVI is a potential sorbent used for the removal and immobilization of heavy metals from contaminated soils and waters. 

### 3.2. Sorption Isotherms, Removal Efficiency from Water, Removal Mechanism, and Release of Ni^+2^

The sorption isotherm shows the relationship between the Ni^+2^ concentration at equilibrium and the amount of Ni^+2^ sorbed on ZVI nanoparticles [18]. In this study, the sorption isotherm was used to determine the sorption capacity of nZVI particles to Ni with different nZVI application rates of 1, 5, or 10 g/kg in the soils, which contained Ni concentrations initially ranging from 50 to 800 mg Ni^+2^/L (Figure 1A). The results showed that the nanoscale zero-valent iron particles significantly alternated the Ni^+2^ sorption isotherm in the sandy loam soils amended with biosolids (Figure 1A). Different Ni^+2^ sorption capacities were observed among different nZVI treatment levels of 0, 1, 5, or 10 g/kg soil. The initial Ni^+2^ concentration in the soil significantly affected the Ni^+2^ sorption to the soil that was amended with biosolids and treated with nZVI particles. The amount of Ni^+2^ sorbed per unit mass of the nZVI-treated soil was increased at room temperature with the increase of initial Ni^+2^ concentration from 50 to 800 mg/L. Increasing the initial soil Ni^+2^ concentration would allow more Ni^+2^ molecules in the soil solution to be sorbed on nanoparticle surfaces in the soil. Thus, Ni sorption in soil is highly dependent on the initial Ni^+2^ concentration. At a lower initial Ni concentration, Ni^+2^ mass in soil solution would be fairly low relative to the amount of available active sites for sorption [25] if, without any nZVI treatments, the soils with or without biosolids amendment showed moderate affinity for Ni^+2^ sorption. However, with the nZVI treatments, Isotherm of Ni sorption was altered from C- to H-type (Figure 1A).

In this study, the H-type isotherm became dominant at all nZVI treatment levels. Interestingly, the H-type sorption isotherm in nZVI-treated soil reflects significant interactions between Ni and nZVI-treated soil, while the C-type isotherm in biosolids-amended soil without nZVI treatments indicates intermediate affinity for Ni sorption. Therefore, the quantity of sorbed Ni by the soil-mix followed a descending order of 10 g/kg nZVI and biosolids-treated soil > 5 g/kg nZVI and biosolid-treated soil > 1 g/kg nZVI and biosolids-treated soil > biosolids-treated soil (control, with no nZVI treatment) (Figure 1A).

It was speculated that the nZVI treatments could contribute many more sorption sites for ZVI nanoparticles, as well as iron hydroxides in the soils. The high specific surface area of nZVI and amorphous iron hydroxides and additional negative charges from biosolids amendment could significantly increase the cation exchange capacity (CEC) in the soil (Table 1).

At a solution containing Ni concentration of 800 mg /L, the amount of sorbed Ni in 5% biosolids-amended soil was 7500, 73,245, 75,022, and 76,888 mg Ni/kg when the soil was treated with nZVI of 0, 1, 5, and 10 g/kg soil, respectively (Figure 1A). High specific surface areas of nZVI significantly enhanced the Ni sorption due to nZVI particles and the amorphous nature of metal hydroxides present in nZVI. Dongsheng et al. [30] reported that nZVI are effective in removing metal ions, such as As(V), As(III), Cd(II), Cr(VI), Cu(II), and Ni(II) from contaminated water and soils. The chemical reduction of Ni^+2^ to Ni occurred due to a slightly higher standard potential of Ni than that of Fe. During the reduction process, metallic iron will serve as an electron donor and provide electrons to reduce Ni (II) [28,29].
Fe^0^ → Fe^2+^ + 2e-; Ni^+2^ + Fe^0^ → Ni^0^ + Fe^2+^

Increasing the nZVI treatment level from 1g/kg to 10 g/kg would also increase the efficiency of Ni retention in the soil (Figure 1B).

The results show that, at a Ni^+2^ concentration of 50 mg/L, the Ni^+2^ removal efficiency from solution increased from 71.3% to a maximum 96.2% when the nZVI treatment level increased from 1 to 10 g/kg. However, at 800 mg Ni^+2^/L, the efficiency increased from 91.6% to 96.1% with the increase of nZVI treatment level from 1 to 10 g/kg. Further, a similar efficiency was also observed at initial Ni concentrations of 100, 200, and 400 mg/L (Figure 1B). The increased adsorption or removal efficiency from solutions with increasing nZVI treatment levels was mainly due to the addition of more active sorbing sites that retain more Ni^+2^ ions on their surfaces [31].

Figure 1C shows the effect of pH on the adsorption or removal efficiency of Ni^+2^ from solution by soil–biosolids–nZVI mix. The efficiency of Ni^+2^removal increased with increasing pH from 5 to 11 [32] and reached 99.56% at pH 11 with the nZVI treatment of 10 g /kg. The initial pH significantly affects the Ni removal efficiency [33] because of more OH- at higher pH. More negative sites on nZVI surfaces and increased electrostatic attraction between nZVI and Ni^2+^ will also consequently increase the Ni^+2^ removal efficiency [34]. The competition between H^+^ and Ni^2+^ ions, which reduces the amount of Ni sorbed and reduces the Ni^+2^ removal efficiency, was the reason for decreasing Ni^+2^ removal efficiency at lower pH values.

### 3.3. Iron Nanoparticles as a Reductant and Sorbent

Because of its moderately low oxidation-reduction potential, zero-valent iron has long been used as an inexpensive, effective and environment-friendly reductant [35,36].
Fe → Fe^2+^ + 2e-  E^0^ = −0.41

In addition to oxygen and water, common electron acceptors, many contaminants can serve as the ultimate electron acceptors. In aqueous solutions, iron can react with water and forms a layer of oxy-hydroxide on the particle surface (FeOOH). Its sorption behavior in aquatic environments has previously been reported [37,38,39]. Iron oxides can have metal- or ligand-like coordination properties relying upon the chemistry of solution. The solution pH plays an important role in the sorption process. When solution pH is below the isoelectric point (8.1–8.2), iron oxides are +ve charged and attract -ve ligands. While the solution pH is above the isoelectric point, the oxide surface will become -ve charged and can form surface complexes with cations, including Ni (II). The addition of iron nanoparticles of over 0.01 g/L in wastewater will increase water pH to the range of 8–10 and enhance Ni sorption and removal from wastewater.

The Fourier transmission infrared spectroscopy (FTIR) analysis was performed to investigate the mechanism of Ni^+2^ retention in soil after addition of the nZVI to the biosolids-amended soil (data not shown). The FTIR spectra of the nZVI-biosolids-amended soil showed stretching and bending of HOH, as well as bending vibration of hydroxyl groups on metal oxides; disappearing and shifting of some peaks was noticed. These observations suggest that chemical reactions occurred between Ni^+2^ and OH groups on nZVI particle surfaces.

Trials were also conducted in this study to determine the desorption rate of Ni^+2^ sorbed on biosolids-amended soils treated with nZVI at both 50 and 800 mg Ni/L (Figure 2). The Ni desorption rate decreased from 92 to 7, 4, and 1% with the increase of the nZVI levels from 0 to 1, 5, and 10 g/kg, respectively, at an initial Ni^+2^ concentration of 50 mg/L. When the initial Ni^+2^ concentration increased to 800 mg/L, the Ni desorption rate decreased from 98 to 11, 9, and 6% with increase in the nZVI treatment level from 0 to 1, 5, and 10 g/kg, respectively (Figure 2). A low to moderate Ni desorption rate indicates the stability of Ni^+2^ bound to the biosolids-amended soils that were also treated with nZVI particles.

### 3.4. Sorption Isotherm Modeling

Table 2 and Figure 3A describe Ni sorption on nZVI-treated and biosolids-amended soil by the Langmuir isotherm model. The R^2^ and SE values of sorbed Ni^+2^ described by the Langmuir model were 0.98-0.98-0.92-0.98 and 0.0002-1.62 × 10^−5^-2.94 × 10^−5^-1.23 × 10^−5^, respectively at 0, 1, 5, and 10 g/kg nZVI-treated soil (Figure 3A; Table 2). The SE values for Ni^+2^ sorption calculated by Langmuir in the soil treated with nZVI and amended with biosolids were small. Therefore, the Langmuir model was the best-fitting to the experimental data, suggesting that the Ni^+2^ removal from the surface of nZVI-biosolids-amended soils was in a monolayer mode of adsorption.

As shown in Table 2, The maximum adsorption capacity (𝑞_max_) of 10 g/kg nZVI-treated soil was 125 mg/g, which was much higher than those from the other treatments of 0 (5 mg/g), 1 (25 mg/g), and 5 g/kg (33 mg/g). Indeed, the specific surface area and the nZVI treatment in biosolids-amended soil increases sorption sites on soil surface. High specific surface area of nanoparticles greatly enhanced the adsorption capacity and surface reactivity of nZVI-treated soil [40].

The Freundlich isotherm model was also used to describe Ni^+2^ sorption in the biosolids-amended soil treated with nZVI (Table 2 and Figure 3B). Nickel sorption fitted well to the Freundlich equation for all treatments with R^2^ ranging from 0.87 to 0.99. Freundlich parameters were calculated from the slope and intercept of the linear equation in Table 2. A larger K_F_ value with nZVI treatment reflects a higher Ni sorption capacity of the nZVI-biosolids-treated soil [41]. Table 2 shows that the highest Freundlich K_F_ value (2757.28 mL/g) was observed with the nZVI treatment of 10 g/kg but the lowest (3.69 mL/g) with 0 g/kg nZVI treatment. Clearly, increasing nZVI treatment levels added to biosolids-treated soil greatly increased its capability for Ni^+2^ adsorption or removal from aqueous solutions. The 1/n values of Freundlich isotherm were within a narrow range (1.03–1.71) for the three studied nZVI dosages, in addition to control treatment, and reflect the observed similarities of the overall shape of Ni isotherms, as shown in Figure 3B. In this study, the Freundlich model successfully predicted Ni^+2^ sorption on soil–biosolids–nZVI mix at a concentration range of 50–800 mg Ni^+2^/L of solution. However, the SE values calculated using the Freundlich model for Ni^+2^ sorption in the biosolids- and nZVI-treated soil were much higher than those of the Langmuir model, which indicates the superiority of the Langmuir model over the Freundlich model in describing Ni^+2^ sorption in the soil treated with biosolids and nZVI.

### 3.5. Effect of Reaction Time and Kinetic Models Fitting

The sorption capability of nZVI as sorbent and determination of the time required for Ni^2+^ sorption equilibrium in the soil treated with biosolids and nZVI were assessed by performing kinetic experiments. The Ni^+2^ sorption or removal efficiency versus reaction time is shown in Figure 4A. Clearly, the sorption or removal efficiency of Ni (II) by nZVI applied to biosolids-amended soil increased gradually as the incubation time increased to 120 min and then tended to become stable thereafter.

For 800 mg/L Ni^+2^ solutions, the maximum Ni^+2^ removed by nZVI/biosolids-treated soil were 73,245, 75,022, and 76,888 mg/kg at 1, 5, and 10 g nZVI/kg, respectively, as the equilibrium was reached after 24 hrs (Figure 4A).

The sorption kinetics of Ni^+2^ on nZVI sorbent exhibited an immediate fast sorption, by which about 95–99% of Ni^+2^ was sorbed in the first 30 minutes at all nZVI treatments to biosolid-amended soil and followed by a slow sorption process at 298 K (Figure 4A). The fast sorption of Ni^+2^ during the first 30 minutes was due to saturation of nZVI surfaces with Ni^+2^ ions and fast filling up of active sites in the first phase, followed by a slow sorption phase because of the reduction of active sites on nZVI.

To investigate the sorption mechanism of Ni(II) by nZVI, first-order [42] and power function [43] kinetic models were further applied to fit experimental data. Table 3 and Figure 4B,C showed the conformity of experimental and expected data through R^2^ and SE values. Higher R^2^ and lower SE values indicated best fitting of model and success in describing the Ni^+2^ sorption kinetics on nZVI sorbent. The function of each tested model was different. In the first-order model, the “ln q” vs “t” relationship was linear if the conformity of sorption–desorption to first order equation was found. Additionally, in the power function model, the parameters k_a_, C_o_, and 1/m were obtained from the intercept and slope.

**Table 3 materials-14-02655-t003:** First order and Power function kinetic model parameters for Ni^+2^ sorption by biosolids- and/or nZVI-treated soil of 50–800 mg Ni/L of solution at 25 °C.

Models	Parameter	Biosolids-Treated Soil	nZVI Added to Biosolids-Treated Soil, g/kg		
			1	5	10
First-order Ln (q_o_ − q) = a − k_a_ t [42]	K_a_ µg g ^−1^ min^−1^	0.971	1.253	1.023	0.832
	a µg g ^−1^	10.05	10.17	9.54	7.43
	R^2^	0.92	0.98	0.97	0.96
	SE	0.665	1.490	1.095	1.203
Power Function q = k_a_C_o_ t^1/m^ [43]	Ka min^−1^	2439.49	68,092.61	70,778.28	75,840.29
	1/m	0.176	0.012	0.009	0.002
	R^2^	0.85	0.80	0.79	0.72
	SE	0.065	0.005	0.004	0.001

q = the amount of sorbed Ni^+2^per gram of sorbent in time t; K_a_ = coefficient of sorption diffusion rate (μg/ g min); C_o_ = initial Ni^+2^concentration (mg/ L); t = reaction time (min); k_a_ = coefficient of sorption rate (min^-1^); 1/m = a constant; q_o_ = amount of sorbed Ni^+2^ at equilibrium.

### 3.6. Effect of nZVI on Nickel Fractions in Biosolid-Amended Soil

Figure 5 shows the changes in nickel fraction percentages in biosolids-amended soil before and after treatments of nZVI levels at normal pH (7.73). The nickel fractions of control soil was significantly different from those in the biosolid- and/or nZVI-treated soils. In control soil, the nickel fraction percentages were: exchangeable 3.1%, carbonate-bound 4.44%, Fe–Mn oxides-bound 8.94%, organic matter-bound 7.12%, and residual 76.40%. In the biosolids-amended soil, exchangeable was 7.33%, carbonate-bound was 9.66%, Fe–Mn oxides-bound was 12.5%, organic matter-bound was 30.88%, and residual was 42.63%. However, when the nZVI was added to biosolids-amended soil, Ni fractions were significantly affected (Figure 5). The nZVI treatment of 1 g/kg significantly increased the residual fraction of Ni up to 58.91% in comparison with biosolids-treated soil, while other Ni fractions decreased, except for the Fe–Mn oxide-bound fraction, which increased to 26.13%. Furthermore, increasing the nZVI treatment level from 1 to 10 g/kg drastically increased the stable residual fraction of Ni from 64.58 to 67.66% (Figure 5).

The relatively high percentage of Nickel bound to Fe–Mn oxides in the soil was mainly due to application of nZVI to biosolids-treated soil. Moreover, the increase of residual fraction of the Ni form after addition of nZVI was due to the conversion of all available and moderately-available forms of Ni into unavailable forms of nickel as a residual fraction (reduced, immobilized, and sorbed forms). The bioavailability of nickel in the 5 % biosolids-amended soil treated with different levels of nZVI followed the order of: control soil without nZVI treatment > 1 g/kg nZVI-treated soil > 5 g/kg nZVI-treated soil > 10 g/kg nZVI-treated soil (Figure 5). These results were generally in accordance with other previous observations that showed the great immobilization and removal capacity of nZVI for different heavy metal ions in both soil and groundwater [27]. The accepted mechanism of nZVI for heavy metal immobilization in an aqueous system is a sorption process, reduction of metal ion to zero-valent metal, and (co) precipitation [27,44].

### 3.7. Effect of nZVI on the Mobility of Nickel through Packed-Column

For the evaluation of sorption process, the flow rate is an important parameter [45]. Therefore, the flow rate effect on sorption of Ni^+2^ by nZVI was studied. The flow rate of nickel-contaminated solution was 0.5 mL/min and the Ni^+2^ concentration in the solution was kept at 50 mg/L (Figure 6). The removal efficiencies of Ni by nZVI after five pore volumes under continuous flow conditions using packed-column were 14.8, 28, 95.8, 97, and 99.6% for control soil, soil with biosolids amendment, nZVI treatments of 1 g/kg, 5 g/kg, and 10 g/kg (Figure 6). Moreover, increasing nZVI contents in biosolids-amended soil significantly decreased Ni concentrations in the first pore volume from 55 to 0.5 mg/L in comparison with control soil and decreased from 45 to 0.33, from 40 to 0.11, from 38 to 0.08, and from 35 to 0.04 mg/L in the second, third, fourth, and fifth pore volume, respectively (Figure 6). Therefore, this study demonstrated that nZVI can be used as a good sorbent for the removal of Ni^+2^ from wastewater. These results are in accordance with previous research findings [46].

## 4. Conclusions

The results of this study revealed that there were big differences in Ni^+2^ sorption capacity between the studied levels of nZVI. The initial Ni^+2^ concentration significantly affected the Ni^+2^ sorption on nZVI-biosolids-treated soil mix. It is interesting to note that the sorption isotherm shape “H” of nZVI-treated soil reflects strong interaction between Ni and the nZVI-treated soil while the shape “C” of biosolids-treated soil without nZVI application indicates intermediate affinity for Ni^+2^ sorption. The increase of nZVI levels from 1 to 10 g/kg led to the increase of efficiency of Ni^+2^ retention in the soil. The removal efficiency of Ni(II) by nZVI from solution was increased with increasing pH in the range of 5 to 11 and reached a maximum of 99.56% at pH 11 and nZVI treatment of 10 g/kg. The Ni desorption rate from soil–biosolids–nZVI mix loaded with Ni was decreased from 92 to 7, 4, and 1% with increasing nZVI content from 0 to 1, 5, and 10 g/kg at an initial Ni concentration of 50 mg/L. The maximum adsorption capacity (𝑞_max_) value of 10 g/kg nZVI-treated soil (125,000 µg/g) was much higher than those of 0, 1, and 5 g/kg (5000, 25,000, and 33,000 µg/g, respectively). The possible mechanism underlying nZVI-immobilization of heavy metal ions in an aqueous system is controlled by sorption, reduction of metal ion to zero-valent metal, and (co)precipitation. Moreover, increase of nZVI content to biosolids-amended soil significantly decreased Ni^+2^ concentrations and mobility in the packed-column in the five pore volumes. Therefore, it is suggested that the obtained nZVI could be used as a good sorbent for the removal of Ni^+2^ from wastewater. It appears that iron nanoparticles are effective for the immobilization of nickel present in biosolids -treated soil or Ni^+2^-contaminated aqueous solutions and potentially have advantages over conventional methods for remediation or stabilization of heavy metals in contaminated soils. More significantly, iron nanoparticles can effectively reduce and immobilize nickel in contaminated soil due to their extremely small particle size and penetration into the intra-aggregate pores of contaminated soil. Moreover, the end products of iron nanoparticle reactions are iron hydroxides and oxides, similar to the iron minerals ubiquitous in the natural environment. The addition of small amounts of iron nanoparticles may, in fact, enhance the nutritional value of biosolids for land applications. The potential of iron nanoparticles for the remediation of biosolids-contaminated soil merits further investigation.

## Figures and Tables

**Figure 1 materials-14-02655-f001:**
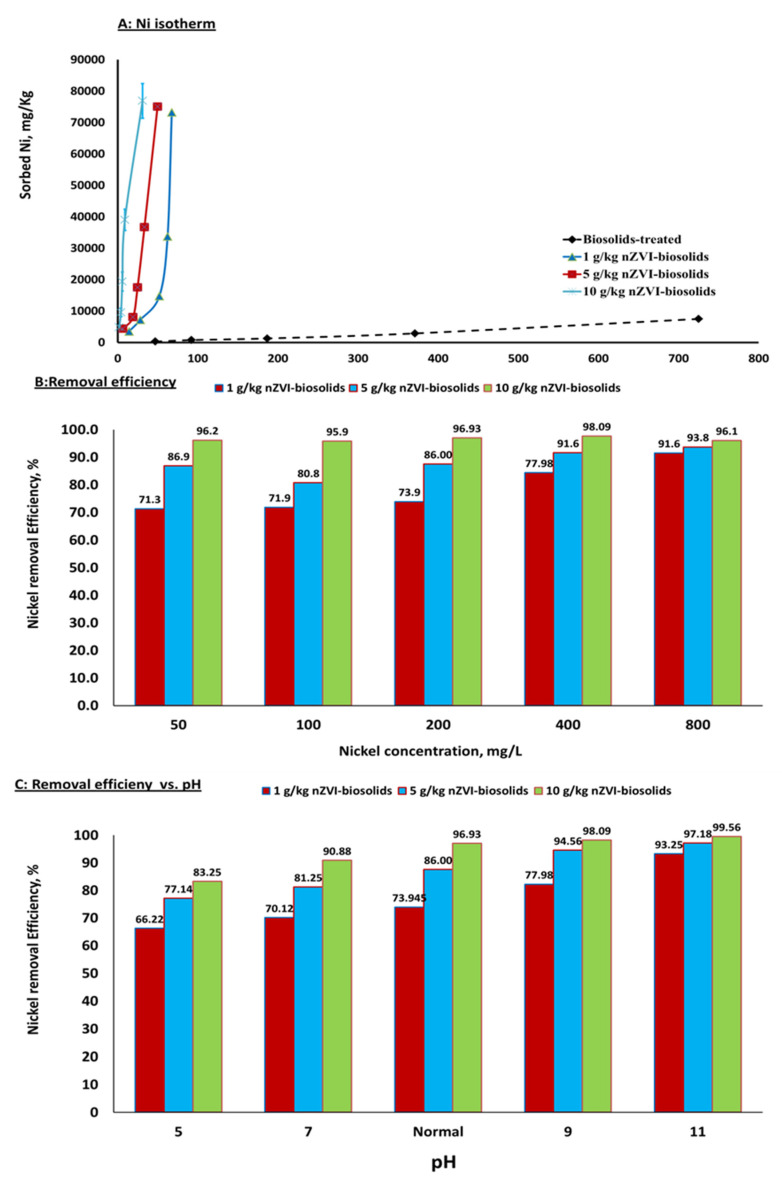
Effect of initial Ni^+2^ concentration on quantity of sorbed Ni^+2^ (**A**) and Ni^+2^ Ni removal efficiency from solution (**B**). (**C**) effect of pH on Ni^+2^ removal efficiency from solution [nZVI dosage = 1, 5, and 10 g/kg; pH = 5–11 ± 0.1; t = 25 ± 2 °C; reaction time, 24 h; shaking speed, 300 rpm]. Error bars represent relative standard deviations (n = 3).

**Figure 2 materials-14-02655-f002:**
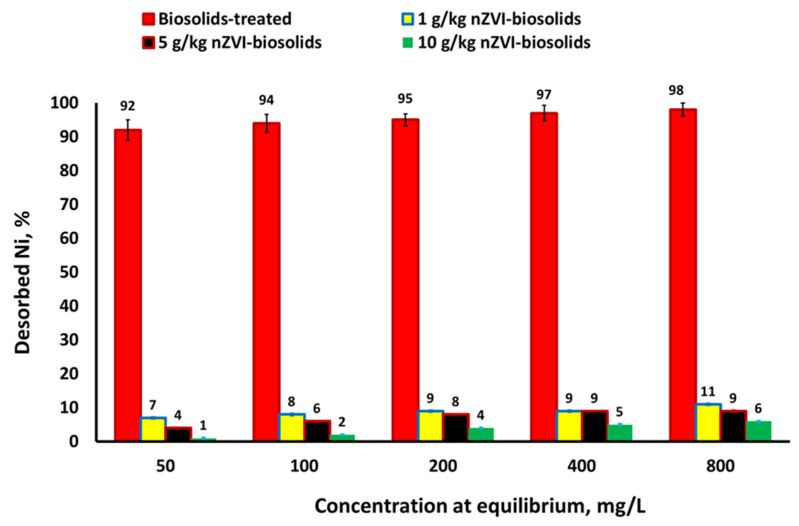
Effect of initial concentration on quantity of released Ni^+2^ from soil–biosolids–nZVI mix loaded with Ni^+2^ [nZVI dosage = 1, 5, and 10 g/kg; pH = 7.73 ± 0.1; T = 25 ± 2 °C; reaction time, 24 h; shaking speed, 300 rpm]. Error bars represent relative standard deviations (n = 3).

**Figure 3 materials-14-02655-f003:**
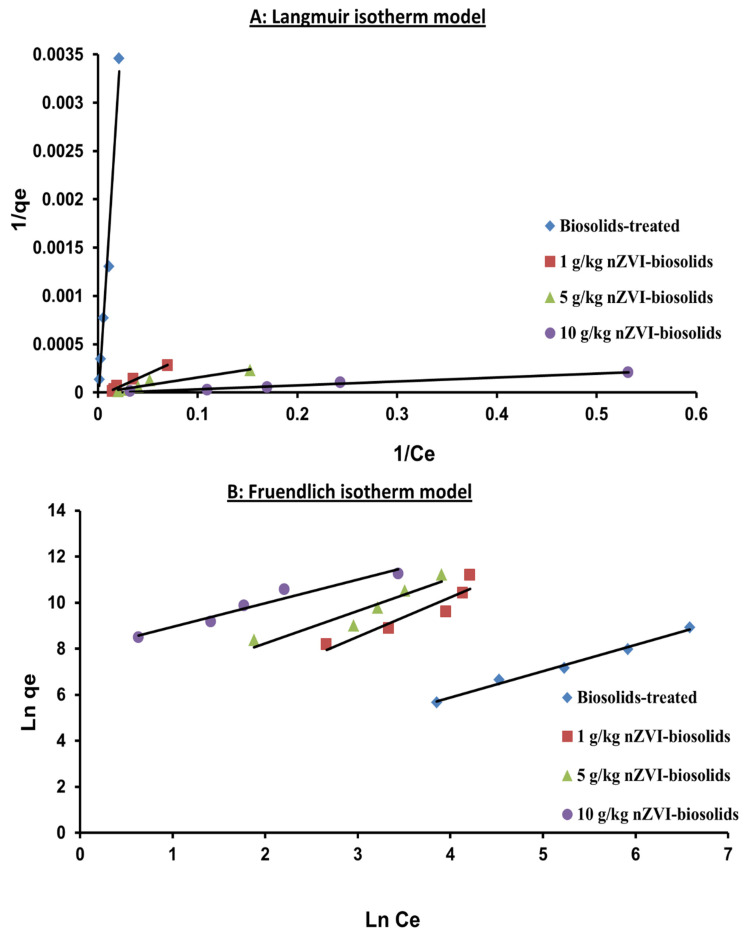
Langmuir (**A**) and Freundlich (**B**) isotherm models for Ni^+2^ concentrations in biosolids-treated soil amended with different levels of nZVI [nZVI dosage = 1, 5, and 10 g/kg; pH = 7.73± 0.1; T = 25 ± 2 °C; reaction time, 24 h; shaking speed, 300 rpm]. Error bars represent relative standard deviations (n = 3).

**Figure 4 materials-14-02655-f004:**
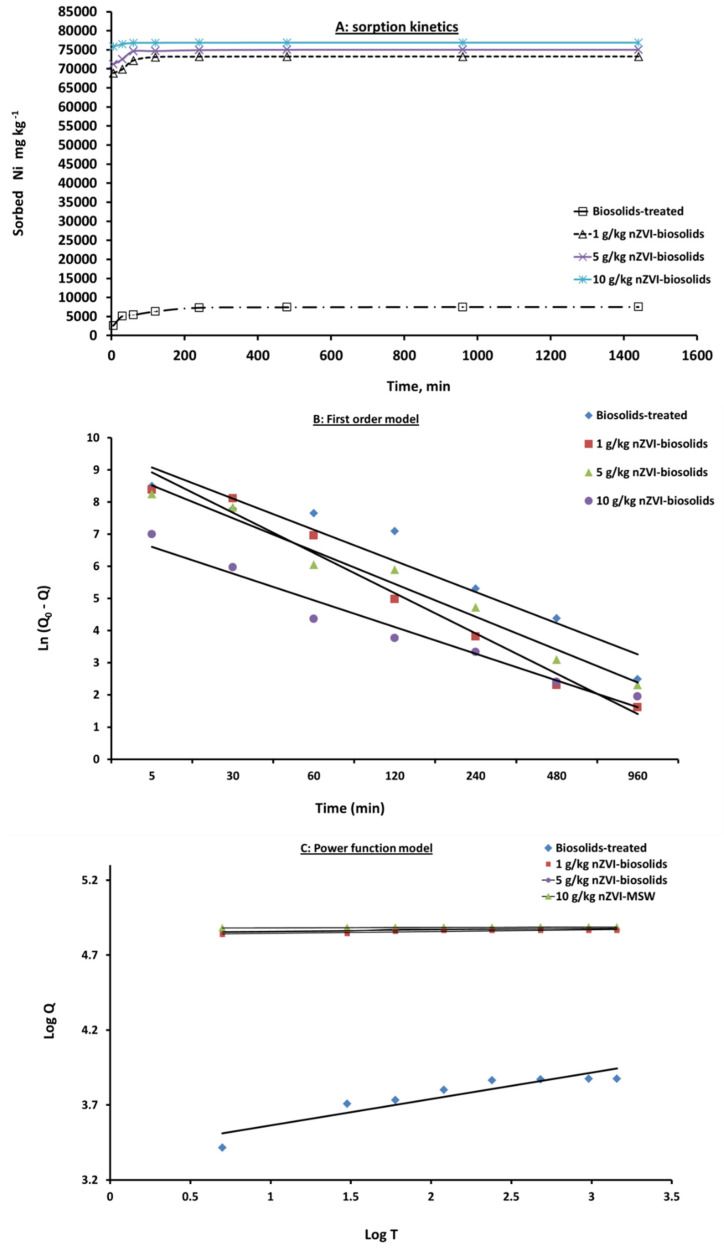
Kinetic of Ni^+2^ sorbed (**A**), first-order model (**B**), and power function model (**C**) in biosolids-treated soil amended with different levels of nZVI [nZVI dosage = 1, 5, and 10 g/kg; pH = 7.73 ± 0.1; T = 25 ± 2 °C; reaction time, 24 h; shaking speed, 300 rpm]. Error bars represent relative standard deviations (n = 3).

**Figure 5 materials-14-02655-f005:**
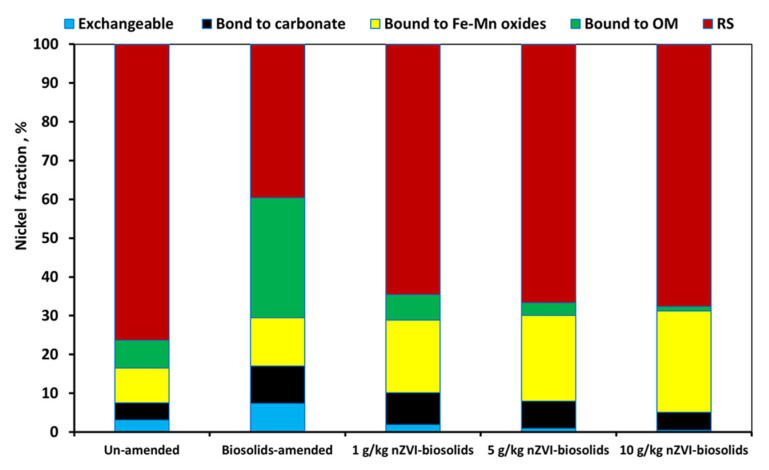
Nickel fractions in biosolid-treated soil amended with different levels of nZVI [nZVI dosage = 1, 5, and 10 g/kg; pH = 7.73 ± 0.1; T = 25 ± 2 °C; reaction time, 24 h; shaking speed, 300 rpm]. Error bars represent relative standard deviations (n = 3).

**Figure 6 materials-14-02655-f006:**
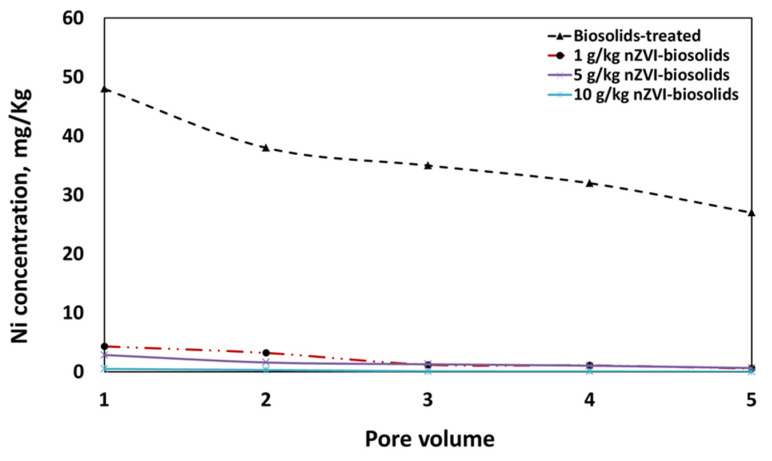
Ni concentration in different pore volumes of biosolids-treated soil amended with different levels of nZVI [nZVI dosage = 1, 5, and 10 g/kg; pH = 7.73 ± 0.1; T = 25 ± 2 °C; reaction time, 24 h; shaking speed, 300 rpm]. Error bars represent relative standard deviations (n = 3).

**Table 1 materials-14-02655-t001:** Some physical and chemical characteristics of the soil and biosolids used in the study. Values are means ± standard deviation (*n* = 3).

Characteristics	Soil	Biosolids
pH *	7.75 ± 0.06	6.38 ± 0.05
Electrical Conductivity (EC) (dS/m)	2.13 ± 0.04	8.33 ± 0.17
Organic Matter (g/kg)	3.80 ± 0.18	410.00 ± 3.77
Cation Exchange Capacity (CEC) (cmol (+)/kg)	12.00 ± 1.07	77.77 ± 2.58
Sand (g/kg)	738.00 ± 3.70	N/A^†^
Silt (g/kg)	106.40 ± 1.90	N/A
Clay (g/kg)	155.60 ± 3.20	N/A
Soil Texture	Sandy loam	N/A
CaCO3 (g/kg)	33.20 ± 3.69	N/A
Total Ni (mg/kg)	13.50 ± 0.11	6.23 ± 0.58
Exchangeable Ni (mg/kg)	6.07 ± 0.08	0.16 ± 0.06
Soluble Ni (mg/kg)	0.24 ± 0.01	0.09 ± 0.02

***** pH measured in sample/water suspension (1:2.5) by pH-meter instrument (CRISON); ^†^: Not applicable; EC: electrical conductivity; O.M: organic matter; CEC: cation exchange capacity; S.L.: sandy loam.

**Table 2 materials-14-02655-t002:** Freundlich and Langmuir isotherm model parameters for Ni^+2^ sorption on biosolids- and/or nZVI-treated soil of 50–800 mg Ni^+2^/L of solution at 25° C.

Models	Parameter	Biosolids-Treated Soil	nZVI Added to Biosolids-Treated Soil, g/kg		
			1	5	10
Freundlich q_e_ = K_F_C_e_^1/n^	*K_F_* (mL g^−1^)	3.69	30.32	228.69	2757.28
	1/n	1.14	1.71	1.4	1.03
	R^2^	0.99	0.87	0.89	0.94
	SE	0.14	0.51	0.44	0.29
Langmuir q_e_ = q_max_(K_L_ C_e_/1 + K_L_C_e_)	q_max_ (μgg^1^)	5000	25,000	33,333.30	125,000
	K_L_ (L mg^−1^)	3.2 × 10^−5^	1.88 × 10^−7^	4.80 × 10^−9^	3.2 × 10^−9^
	R^2^	0.98	0.98	0.92	0.98
	SE	0.0002	1.62 × 10^−5^	2.94 × 10^−5^	1.23 × 10^−5^

q_e_ (mg g^−1^) is Ni adsorbed per gram of adsorbent, C_e_ (mg L^−1^) is equilibrium Ni^+2^ concentration in solution, K_f_ is a constant related to adsorption capacity of the adsorbent (Lmg^−1^), n is a constant, q_max_ (mg g^−1^) is the maximum adsorption capacity of the adsorbent, K_L_ (Lmg^−1^) is Langmuir constant related to the free energy of adsorption.

## Data Availability

The data presented in this study are available on request from the corresponding author.

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
