# Peer review of "Zero-Valent Iron Nanoparticles Remediate Nickel-Contaminated Aqueous Solutions and Biosolids-Amended Agricultural Soil"

_materials, 2021, doi:10.3390/ma14102655_

Round 1

Reviewer 1 Report

The subject remains timely. Attention to the following aspects will improve the contribution:

1. Fundamentally, the batch kinetics are expected to be zero or first order, without seeking a more complex, polynomial fit. Consider the chemical engineering of the reactor system. This enables theoretical performance curves to be drawn, stating assumptions then the data points can be compared to model predictions.

2. Fluid flow and mass transfer need to be considered in a more detailed appraisal of reaction environment.

3. The active area needs to be expressed and monitored with time.

4. Lifetime and ability to regenerate and cycle the materials are unclear.

5. It is useful to calculate the normalised space time yield for 90% contaminant removal. This is the volume of waste in which the contaminant concentration can be reduced by 90% in a unit volume reactor in unit time, units: m3 m-3 h-1.

Reviewer 2 Report

The paper "Zero-valent iron nanoparticles remediate nickel-contaminated aqueous solutions and biosolids-amended agricultural soil" can be published after minor revision. Some suggestions should be considered:

1) I believe that the introduction should describe the toxicity of nickel(II) to living organisms, humans and include how it migrates into the environment. It is worth citing the paper: Processes 2021, 9(2), 285; https://doi.org/10.3390/pr9020285

2) The notation of nickel ions as Ni(II) should be standardized throughout the paper. Nickel in the metallic state is not adsorbed, but nickel ions on the +2 oxidation state are.

3) The determined sorption capacity values should also be recorded in the paper and expressed in the same units.

4) In my opinion, 5 experimental points for the preparation of the isotherms is too little, why solutions with the higher concentrations of Ni(II) ions were not made.

Translated with www.DeepL.com/Translator (free version)
